

# Symmetry meets AI

**Gabriela Barenboim[1], Johannes Hirn[1] and Verónica Sanz[1,2]**

**1** Departament de Física Teòrica and IFIC,
Universitat de València-CSIC, E-46100, Burjassot, Spain
**2** Department of Physics and Astronomy, University of Sussex, Brighton BN1 9QH, UK

## Abstract

We explore whether Neural Networks (NNs) can *discover* the presence of symmetries as they learn to perform a task. For this, we train hundreds of NNs on a *decoy task* based on well-controlled Physics templates, where no information on symmetry is provided. We use the output from the last hidden layer of all these NNs, projected to fewer dimensions, as the input for a symmetry classification task, and show that information on symmetry had indeed been identified by the original NN without guidance. As an interdisciplinary application of this procedure, we identify the presence and level of symmetry in artistic paintings from different styles such as those of Picasso, Pollock and Van Gogh.


## 1 Introduction

Symmetries are central to the underlying structure of Nature. The discovery of a symmetry signifies the existence of a fundamental principle and manifests itself in the form of physical laws and selection rules. Indeed, all known fundamental laws of Physics can be derived from an axiom of invariance under a transformation. This is exemplified in Galilean relativity,

Maxwell's equations for electromagnetism, Einstein's special and general relativity as well as the gauge theories of the fundamental forces in Particle Physics.

On a more pragmatic note, symmetries have lots of applications, such as those in crystallography or the simplifications they confer to the study of a problem: a symmetry is an organizing structure underlying the information at hand. Discovering such a pattern thus leads to a deeper understanding, as in the simple case of a Rohrschach test: noticing the reflection symmetry of an inkblot helps a child guess how the drawings were made, i.e. by folding a blotted paper onto itself.

This understanding allows for simplifications in the way we handle the data and, at a deeper level, can indicate the presence of a higher-level principle. This connection between symmetry and simplicity *or even elegance* appears frequently in Theoretical Physics.

In Art, symmetry is also often linked to the concept of elegance. This is not to say that symmetric artworks are more beautiful, as it is known that most humans prefer faces, musical pieces, paintings and photographs where the symmetry is not exact, but slightly imperfect or broken [1,2]. In Physics as well, deviations around a symmetric situations are often considered as a useful approximation technique, since perfect symmetries are seldom found in Nature.

A Physics example of the the discovery of a symmetry is given by the motion of the planet Mars. Before his death in 1601, the astronomer Tycho Brahe had gathered the most accurate records of its position in the night sky. Within these data was an underlying structure that took many years for Johannes Kepler to tease out in the shape of ellipses [1]. From this simpler representation of the data, Isaac Newton was able to deduce the laws of gravity, which exhibit a central symmetry, no doubt a simpler, deeper and thus more general description of the motion of celestial bodies than the original collection of observations. Fast-forwarding many years, we now understand that Newton's laws can be obtained from imposing a symmetry on an abstract object called the Action.

Our idea in this paper is to lay the foundations for an automated, or artificial intelligence (AI), version of the Kepler intermediate step between Brahe and Newton.

A functional task-oriented implementation of the general concept of AI is called Machine Learning (ML). It involves algorithms that give general prescriptions for computers to progressively approximate (or learn) the appropriate rules to reproduce specific observations. This is in contrast with traditional programs, which lack the level of *expressivity* needed here.

Currently, Science in general and Physics in particular are undergoing a revolution of sorts [3], as the ML methods that have been employed in experimental fields with large datasets are applied to more formal areas and even for symbolic mathematics [4].

ML is indeed particularly good at pattern recognition, and we thus ask the question: as these methods are used to extract information from the data, can they also detect the presence of symmetries in the data they are exposed to? And if they can, do they do so automatically, i.e. do they naturally organize the information according to symmetry patterns?

In this paper we walk the first steps to answer the above questions. Beyond our curiosity and our desire to understand not only the laws of nature but the way ML proceeds, we apply our method to study a deep connection between Physics and Art.

After training algorithms on a Physics-based set-up in Sec. 2, we apply them to artworks in Sec. 3 and evaluate their level of symmetry. Many extensions and applications of this work could be pursued and in Sec. 4 we will discuss some ideas in this direction.

---

[1]In this example, there are small perturbations to the heliocentric potential acting on Mars, due to the presence of other planets: the symmetry is realized only approximately in Nature.

## 2 The Physics template

To train an algorithm that learns about symmetries we need a template to describe them. Physics is an excellent starting point, as the use of symmetries is well understood. We also need to design a situation where the condition of symmetry is clear and controllable, and one can generate many examples with specific symmetries, or lack thereof.

Among the physical situations we could choose from, Mechanics offers a perfect ground for dataset generation. When computing trajectories of objects like stars, rockets or billiard balls, the understanding of symmetry properties of the potential they experience is important to simplify the equations of motion.

The method we will describe is split in two parts, a *Decoy Task* and the training of a *CNN Classification Task*.

### 2.1 The decoy classification: binary classification of points

The flow of the first task, which we call *decoy*, is schematically shown in Fig. 1. Along the text we will explain each of the steps in the Decoy Task and refer back to this figure.

**PART I: DECOY TASK**

Figure 1: Part I Decoy Task. Schematic view of the main steps of this task: *1.)* potential definition, *2.)* representation of the equipotential lines (decoy image), *3.)* image processing to be fed into *4.)* a fully-connected neural network and finally, *5.)* extraction of the output of the last hidden layer to perform Principal Component Analysis (PCA).

This task starts by defining potentials with known symmetry properties. For simplicity, we will consider two-dimensional (2D) potentials, $V(x, y)$, as shown in the first step of Fig. 1. We



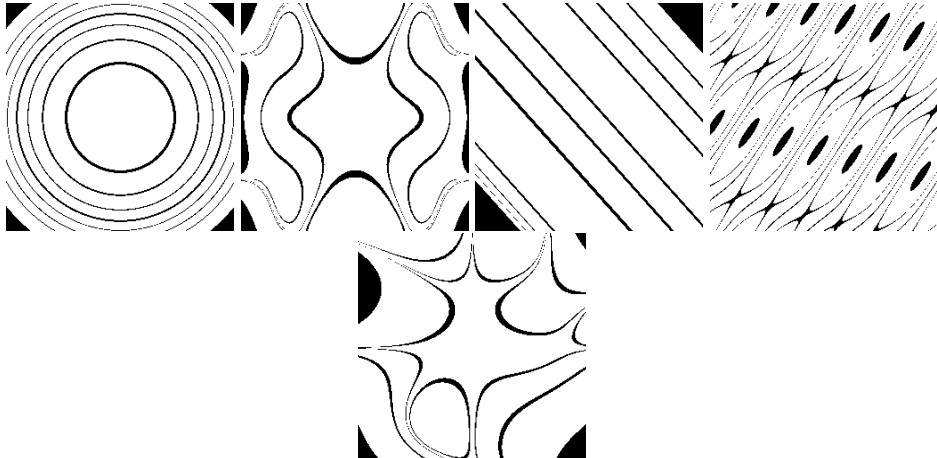

Figure 2: From left to right and top to bottom: Examples of potentials with symmetries ($O(2)$, $Z_2$, $T$, $T_n$, $\emptyset$).

then plot regions in $(x, y)$ with equal values of the potential at given intervals for $V$. The result is shown in the second step in Fig. 1 where we have chosen 5 equipotentials, which result in 5 contours or more. The image of equipotential lines could be simply viewed as altitude contours drawn on a map, albeit a pixelated map of 256 pixels on each side, and serves as a pictorial representation of the symmetry in $V(x, y)$. For example, for the case $V(x, y) = x^2 + y^2$, the value of the potential is the same as we transform any point $(x, y)$ into a new point $(x', y')$ using a rotation matrix of arbitrary angle $\theta$. Intuitively, it is clear that this image will remain the same as we fix the centre and rotate around it.

Apart from rotation symmetry, denoted by $O(2)$, we will also consider other types of 2D symmetries: reflections ($Z_2$), translations ($T$) and discrete translations ($T_n$). Examples of these situations are shown in the first two rows in Fig. 2.

To learn to identify symmetries requires to learn to identify situations where *no* symmetry is present. We will denote this situation by $\emptyset$, and show an example in the last row of Fig. 2.

These image of contour lines are then pre-processed before feeding them into a fully-connected neural network (FCNN). The pre-processing is as follows: we first search for edges using a Canny detector [5]. We then repeatedly perform morphological erosions and dilations [6] until we obtain a stable skeleton which we depict in white. This is the first class in our binary classification problem. We define the second class as the eroded version of the background (shown in black). The last erosion step is designed to create a gap of a few pixels across (depicted in gray) between the two classes, separating them by a no-man's land belonging to neither class. This gives a bit of wiggle room for the FCNN to flip its prediction from one class to the other, making its training easier and less dependent on details of the processing. We then randomly select up to 10,000 pixels per class, making sure that the two classes are balanced. We then train hundreds of FCNNs, each on its own binary classification problem.

This is in a sense a decoy task where we do not look for a symmetry pattern, but just a binary classification of points in a space of a given dimensions. This part is similar to Ref. [7], where they also train an FCNN on a task until it has learned it reliably, and then extract the higher-level features that it has learned. In our case, we do not set up a multi-class problem where each class is defined by a different equipotential, as this would not easily apply to Art or any other system where there was no equivalent of equipotential lines.

What we want to find out is whether the FCNN, at some level, organizes the information according to possible symmetries that may be present in the original potential. To study this, the binary classification problems we give our FCNNs are themselves designed to belong to

**PART II: CNN CLASSIFICATION**

Figure 3: Part II: CNN Classification. The PCAs collected in Part I and labelled are used to train an algorithm to classify symmetries.

one of 5 symmetry classes: invariant according to reflection, discrete translation, continuous translation, continuous rotation, or problems without a known symmetry. Note that at this stage we do not provide these labels to the FCNN.

We build an FCNN with 20 layers of 200 neurons each, and train it to classify the points according to whether they belong to the contours or to the background.

At this point, it is important to remember that we are not teaching a CNN to recognize shapes in a full image, but training an FCNN on individual points to be classified in either of two classes: we are feeding the FCNN individual lines of a table of coordinates in no particular order, and asking it to perform a binary classification for each of these pairs of coordinates, i.e. to predict whether a given point or pixel belongs to the contour or to the background. In practice, the pairs of coordinates are fed by batches in random order without regard to their respective positions in the 2D space of the original image, i.e. a random flattening procedure. Only after a whole epoch has the FCNN seen all the points (coordinate pairs) that were selected from the image.

We use `fastai` [8] with a one-cycle policy [9], maximum learning rate of $5 \times 10^{-3}$, batch size of 8,000 (or less if there are fewer points in the input data) using a training/validation split of 80%/20% for 300 epochs.

## 2.2 The actual classification of PCAs according to symmetries

We then turn to the actual problem of identifying which symmetry class the initial task belonged to. We designed a procedure pictorially described in Fig. 3 [2].

To achieve this, we extract the output of the penultimate layer of the FCNN where we expect the FCNN to have organized the data in the most synthetic form. We project the output of the 200 neurons in this layer onto two dimensions according to the most relevant combinations as given by Principal Component Analysis (PCA). In practice, we select the first two components, so that we end with a 2D result for each FCNN, i.e. an image, that we can in turn feed to a convolution neural network (CNN). We then discretize this image to $224 \times 224$ pixels, encoding the two classes as different colors and the number of points in each bin/pixel as the intensity of that color.

We repeat this task hundreds of times, each time training a new FCNN on a new 2D function, with some functions being designed to be symmetric under reflection, continuous rotations, continuous translation and discrete translations.

---

[2]Note that this second part of our procedure is very different from the proposal laid out in Ref. [7].

Inspecting the resulting PCAs (see Fig. 3), one would be hard pressed to see any difference between PCAs from problems of different symmetry classes. Actually, to the untrained eye, there seems to be just as much variability between PCAs from the same symmetry class, or even PCAs from different training runs of the same FCNN architecture on the exact same 2D potential, as there is between PCAs corresponding to problems with different symmetry classes. The origin of this variability can be traced back to the stochastic nature of training the FCNN, which means that no two PCAs are the same, even for the exact same potential.

Hoping that a CNN would perform better than our eye at distinguishing between PCAs coming from FCNNs trained on different symmetry classes, we then turn to Part 2 of our workflow, which is to train a single CNN to classify hundreds of PCAs. If the CNN succeeds, it means that there is a common pattern in the various PCAs produced by FCNNs trained on different decoy tasks belonging to the same symmetry class. We would therefore conclude that the FCNNs have encoded some information about the symmetry class of the problem it has been learning.

Looking for a good accuracy in our image classification task, we perform transfer learning using a Resnet network, but select the smallest one (i.e. ResNet18 [10]) for speed, as implemented in the `fastai` package [8]. From a training sample of 1240 PCAs, we achieve a validation accuracy of 73% on the 5-class problem, and an 80%-95% accuracy for each of the four different binary problems of identifying the presence or absence of each symmetry separately.

We checked how our results were affected by varying the hyperparameters of the FCNN, in particular as they allow a more precise or looser fit. The most obvious issue is when the FCNNs do not perform so well at their binary classification task: the CNN then has a hard time reaching a good accuracy. For instance, with the same hyperparameters, but only 100 epochs instead of 300, the FCNNs typically reach an accuracy below 99.8% on the whole training plus validation dataset. CNNs trained on the resulting PCAs barely reach 60% accuracy on the 5-class task.

Perhaps less obvious is the fact that it is counterproductive to train the FCNNs until they learn their task perfectly: we have tested this by training FCNNs with the same hyperparameters (including in particular 300 epochs), but without defining a validation set. The best model is then selected by minimizing the (training) loss instead of the validation error as was the case before. Such models typically reach an average error rate below 1/10,000 on the whole dataset, i.e. so small that most of them did not make a single mistake in their binary classification task (involving several thousands of points). Here again, the CNNs trained on the resulting PCAs barely reach 60% accuracy on the 5-class task, possibly because the FCNN has overfit to the location of the individual pixels that have been selected in the random-sampling process instead of relying on simplifying assumptions such as the symmetry.

## 3 Application to Art

Since the very beginning of times, symmetry has been studied not only by scientists but also by artists. Most people are familiar with the broader concept of symmetry. The notions of beauty, proportion, or harmony immediately cross our minds when talking about symmetry, abstract or concrete.

The Merriam Webster Dictionary in its first entry for the term symmetry says: *beauty of form arising from balanced proportions* while the Cambridge dictionary states that symmetry is *the quality of having parts that match each other, especially in a way that is attractive, or similarity of shape or contents*.

Symmetry has been the guiding principle to construct the Physics theories that describe Nature. Even before we developed our first Physics theory, ancient Greeks were captivated by the symmetries of the world around them and believed that these would be reproduced in the underlying principles of Nature itself.

The concept of symmetry is also ubiquitous in the artistic world. The question therefore is whether a concept that crosses scientific and artistic boundaries can be analyzed in the same way by a "blind" observer. Is this rather strict scientific concept correlated with a visual narrative, and does it play a relevant role to the larger appreciation of beauty?

As mentioned in Sec. 2, instead of considering a multiclass problem as in Ref. [7], we trained our FCNNs on a binary class for simplicity. This simplicity also allows us to apply the same binary classification problem to other functions of 2D, like paintings or other physical problems. The guiding idea behind our pre-processing is to approximate the main shapes of the painting as one would do in a felt-tip sketch attempting to convey a few lines of the painting. For this, we first transform the paintings into grayscale images of $256 \times 256$ pixels.

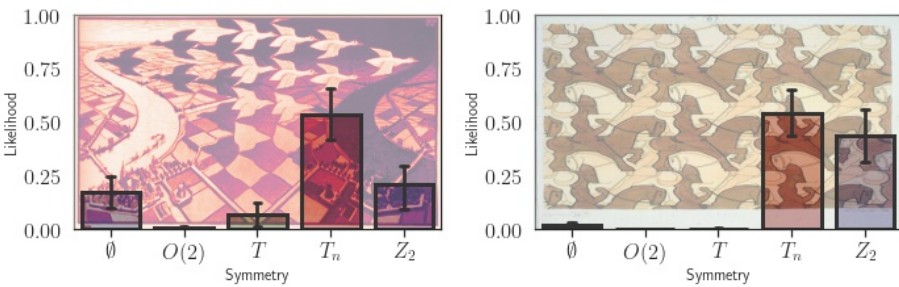

Figure 4: Escher works *Night and day* and *Horseman*. The algorithm's symmetry scores are overlaid on top.

As also explained in Sec. 2, we then use a Canny edge detector to pick up these contours, and then simplify them using successive erosions and dilations until we get a stable skeleton. We then have two classes, namely the contours and their complement: the background. To simplify the problem, we separate the two classes by introducing a no-man's land where they join (a gap of a few pixels belonging to neither class). We do this by performing one morphological erosion on the background.

We move to show various examples of outcomes of the algorithm of symmetry classification on works of Art using the 5-class results. These are illustrative examples of different behaviour, and we have examined a large number of known artistic works, checking that the outcomes are consistent with the approximate symmetries one would expect to find.

We start by analysing the predictions of the algorithm on the drawings of Escher, an artist known for his use of approximate symmetry and optical illusions. In Fig. 4 we depict two drawings, *Night and Day* and *Horses*. We then overlay the AI predictions for each symmetry on top. For each painting, we train the same FCNN architecture several times with the same hyperparameters but using different random initializations of the parameters, yielding different outputs from the hidden layer, and thus different PCAs. We then feed each of these PCAs from the same paintings to the same CNN model we trained on the PCAs of potentials. For each painting, we depict the predicted likelihood for all five symmetries (none $\emptyset$, rotation $O(2)$, continuous translation $T$, discrete translation $T_n$ and reflection $Z_2$) and draw a 68% confidence interval computed via a non-parametric bootstrap on the predictions obtained from running the same CNN on the PCAs resulting from multiple FCNN runs on the same painting.

Both drawings are classified as having discrete translation symmetry and some level of reflection invariance, which they clearly possess. This shows that the algorithm has been able to generalize from the Physics potentials it has been trained on to other, more complex,

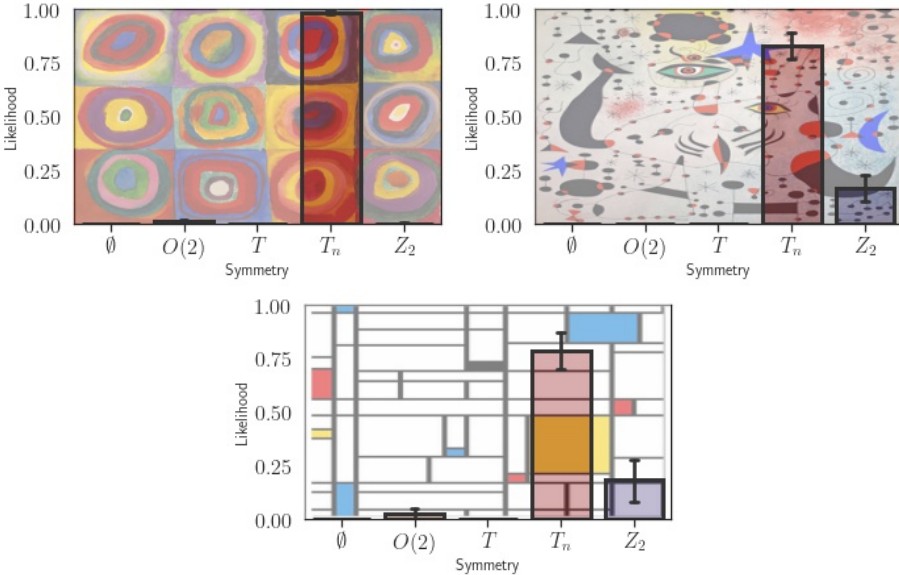

Figure 5: Examples of discrete translation symmetry across artists and schools.

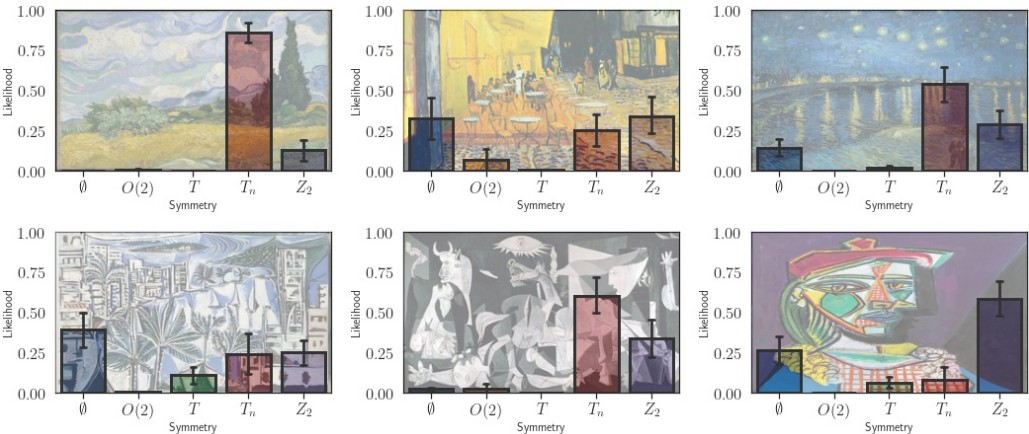

Figure 6: Upper panel Works by Van Gogh: *Wheat Field with Cypresses,Café Terrace at Night, Starry Night Over the Rhône*. *Lower panel* Works by Picasso: *The bay of Cannes*, *Guernica* and *Lover in a beret*. The algorithm's symmetry scores are overlaid on top.

situations without exact symmetries.

Escher drawings are complex and playful, but some of the appeal of artistic works may lie in the repetition of a simple pattern. Examples of such situation are shown in Fig. 5, where works by Kandisnsky, Miró and Mondrian reveal the same dominance of discrete translation invariance.

Across paintings from the same author and period, one can observe differences in the level of symmetry they contain. This is exemplified by Van Gogh's paintings in the upper panel of Fig. 6 and by Picasso's cubist paintings shown in the lower panel in Fig. 6.

In the first painting of Van Gogh, *Wheat Field with Cypresses*, discrete translation is identified, bearing to the painter's repetitive strokes to depict the sky, clouds and ground. Some level of reflection symmetry is also identified, due to the similarity of the ground and foreground overall shapes.

On the second painting in Fig. 6, *Café Terrace at Night*, the same discrete translation sym-

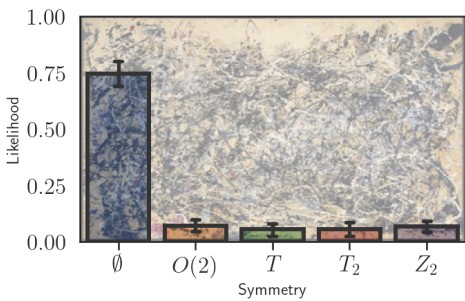

Figure 7: Work by Pollock *Number 1, 1948* and the algorithm output.

metry is found due to the repetition of elements such as tables, now competing with the approximate reflection symmetry clearly present in the painting with the lines of café tables.

The third Van Gogh painting is *Starry Night Over the Rhône*, which exhibits a good amount of reflection and discrete translation symmetry, due to the repetitive patterns of stars and water reflections.

A similar discussion can be drawn from the works by Picasso, where many similar elements, placed at more or less similar distances, would generate an approximate discrete translation invariance. Opposed and similar elements within the painting lead to reflection symmetry, and one can observe that this occurs to a larger amount in the woman's face than in the other two due to the duplication of elements in a human face, even when depicted by a cubist.

So far we have seen examples of artistic works with some degree of symmetry. But artistic manifestations are very diverse, including in the aspect we are analyzing. A clear example of this diversity can be found in a successful artist whose trademark was precisely the lack of symmetry, Jackson Pollock. The algorithm does indeed find in paintings from his *drip period* a dominant non-symmetric result, see Fig. 7, with a negligible amount of all the symmetries.

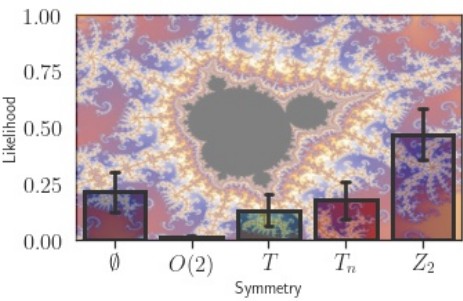

Figure 8: Representation of a Mandelbrot set and the algorithm output.

We would like to finish this section on Art applications with a different type of artistic expression, not human-made. Nature can also create beautiful patterns from complex phenomena, and a well-known display of this *creativity* are fractal images. In Fig. 8 we show a representation of the Mandelbrot set and the output of our algorithm. Despite the intricacies of the image, the algorithm is able to recognize the symmetries a human observer would have identified: discrete translation symmetry due to the repetition of patterns, and invariance under reflection along the axis cutting through the black shape.

# 4 Conclusions and Outlook

Both artists and scientists are human beings trying to decipher the world surrounding them. The skills they develop and the tools they use are different, as different as the audiences they target. But the goals are basically the same: they are both dealing with complexity using the tools at hand, as we all are.

Scientists flourish through sophisticated mathematical theories and perform even more sophisticated experiments trying to break into Nature's hidden secrets. Artists get their messages across going through periods that reflect their vision. The notion of symmetry unifies both worlds, and perhaps a tool to detect symmetry can shed light on both worlds as well.

In this paper we have developed a tool to identify symmetries in 2D representations using Artificial Intelligence methods based on Physics and applied to Art. This work should be understood as a step towards exploring how AI learns about hidden structures, and how this level of abstraction can be put to use.

There are clearly a lot of avenues to explore, and we have just scratched the surface of a field with unprecedented potential: ML techniques can be used not only to learn, but can also teach us what we are missing from our own observations.

To continue exploring, further and more sophisticated algorithms are needed (and are in progress). Towards this end the use of Generative AI techniques [11], with its capacity to learn the deeper *rules of the game*, is showing great potential. Among these tools, Variational Auto-Encoders [12], are specially promising, as well as the generalisation to higher-dimensional representations. Moreover, time-series or issues of self-similarity, as in fractals, could also be handled following similar ideas.

Focusing on Art, one could ask whether the different stages through which an artist evolves can be related to the hidden symmetries in their work, or whether schools and symmetries show any correlation. The ability of AI to discover underlying patterns is likely not restricted to the concept of symmetry and one could pose more advanced questions such as: could an AI learn to identify schools and individual artists' works? Or, could an AI predict the price of a painting?

Turning our attention back to Science and in particular to Particle Physics, one could ask: given the data in Gargamelle [13, 14], the historical heavy liquid bubble chamber detector at CERN, would an AI have discovered the Standard Model, arguably the most successful theory ever? Or even, are we sure the Standard Model is the only possibility? Could a symmetry-trained-CNN analysis reveal something else?

Finally, Art and Physics are not the only areas where a symmetry analysis could lead to new insights. Revealing hidden symmetries in datasets would allow us to understand the system more deeply and perform data augmentation. For example, an analysis of traffic patterns in a city could unveil an approximate symmetry, allowing to model this behaviour creating a better *digital twin* and to apply it to other cities.

# Acknowledgments

We thank Sven Krippendorf and Jose Angel Oteo for discussions. GB and JH acknowledge support from the MEC and FEDER (EC) grant FPA2017-845438 and the Generalitat Valenciana under grant PROMETEOII/2017/033. VS acknowledges support from the UK Science and Technology Facilities Council ST/L000504/1 and the Spanish FPA2017-85985-P. This project has received funding /support from the European Union's Horizon 2020 research and innovation programme under the Marie Skłodowska-Curie grant agreement 860881-HIDDeN.

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
