# Peer review of "Symmetry meets AI"

_SciPost Physics, doi:SciPost Phys. 11, 014 (2021)_

## Round 1 · Referee Report · Tilman Plehn (Referee 1) · 2021-5-7

Report

The paper is very interesting, even though there does not appear all that much physics-specific content. So I recommend it for publication, with a set of comments included in the pdf file (in red). One aspect that could use some more work is the list of references, one of the relevant ML keywords might be `representation learning'.

---

## Round 1 · Referee Report · Anonymous (Referee 2) · 2021-5-10

Report

The authors propose a methodology to extract the underlying mathematical symmetries from the learning procedure of a neural network. They initially train a fully connected neural network (FCNN) with images generated using certain symmetries. Then authors propose a CNN based learning procedure where the initial two principal components collected from FCNN are fed into to classify underlying symmetries. The paper is well-written, relevant for both physics and AI applications, and the results are explained in detail. I would suggest the publication on SciPost if the authors can clarify the following points.

I. In section II.1, the authors are designing a decoy task procedure using FCNN where after a step of preprocessing the template images (fig 2), they are given to an FCNN. This requires a particular procedure of flattening the image. If a large enough network has been given, the flattening procedure should not affect the outcome. I was wondering if the authors have investigated different flattening options or their effects on the outcome. Since this will affect the relationship between neighbouring pixels, I wonder if it affects learning the underlying symmetries.

II. Please avoid this request in case if I missed it in the manuscript. For reproducibility purposes, can authors clarify the specification of the FCNN that has been used? Such as the activation functions, loss functions, if any etc.

III. In section II.2, the authors are using ResNet18 for the classification of the symmetries. I wondered if smaller network options have been investigated and if the reason for choosing such an extensive network is purely for increasing accuracy.

IV. Can the authors clarify the origin of the error bars of the symmetry bins in figures 4 to 8? Does it appear as a result of running the algorithm many times, as described on page 5? Please correct me if I am missing something, but a neural network will result with the same output every time unless it has a Bayesian layer. Is this because the authors are using multiple distorted template images to classify one symmetry; hence, a class is decided by the statistical significance of this collection?

V. I'm aware that it is hard to release a public code, but if it is possible, can authors also release the analysis code that has been used for this study. I'm sure that the community will appreciate and further develop related studies using their code.

---

## Round 1 · Referee Report · Luigi Del Debbio (Referee 3) · 2021-5-14

Report

This manuscript presents a very interesting study of the possibility to use ML to 'detect' symmetries in a given image. The focus here is in understanding how a NN encodes the information about the image that it is being learned, and whether this information allows us to say something about the symmetries of the image. The manuscript should be accepted for publication, the authors may wish to consider some of the remarks below.

The key idea of this work is to analyse the last hidden layer of the NN and determine whether some information about the symmetries of the image is encoded there. The output of the 200 neurons in this layer is projected onto two-dimensions using PCA. The output is an image that is fed into a convolutional neural network that will learn the symmetry classification. The training is done using physics potentials with known symmetry properties, before using the trained machinery onto paintings.

The results are tantalising and should inspire further studies.

I have a few comments that I would like the authors to address.

1) It would useful to specify clearly in the manuscript what is the exact input and what is the output of the FCNN used for the decoy task. What are the details of the training? is there a training set and a validation set?

2) As usual with these kind of studies, it would be good to have some reassurance about the potential systematic errors in the procedure. How strongly do the results depend on tuning some of the hyper-parameters? There are currently a few comments at the end of section II in the manuscript, but it would be useful to expand and provide some more quantitative information. For instance when saying that 'the CNN [...] does not manage to reach the same accuracy...', can this be quantified with a few examples? This point in particular could be important, since the authors suggest that a 'less-perfect' learning would rely on symmetries to encode the image more than a perfect reproduction. This is obviously a very interesting suggestion, which deserves more detailed scrutiny. Surely, if the training is relaxed too much, then informations about the image will be lost. There must be an ideal window where the system works best. Is it possible to explore this feature quantitatively?

Requested changes

I would suggest that the authors consider the remarks above.

---

## Round 2 · Author Response

COMMENTS FROM REFEREES
*It would useful to specify clearly in the manuscript what is the *exact input* and what is the output of the FCNN used for the decoy task. What are the details of the training? is there a training set and a validation set?.
*In section II.1, the authors are designing a decoy task procedure using FCNN where after a step of preprocessing the template images (fig 2), they are given to an FCNN. This requires a particular procedure of flattening the image. If a large enough network has been given, the flattening procedure should not affect the outcome. I was wondering if the authors have investigated different flattening options or their effects on the outcome. Since this will affect the relationship between neighbouring pixels, I wonder if it affects learning the underlying symmetries.
*Please avoid this request in case if I missed it in the manuscript. For reproducibility purposes, can authors clarify the specification of the FCNN that has been used? Such as the activation functions, loss functions, if any etc.
TEXT ADDED
At this point, it is important to remember that we are not teaching a CNN to recognize shapes in a full image, but training an FCNN on individual points to be classified in either of two classes: we are feeding the FCNN individual lines of a table of coordinates in no particular order, and asking it to perform a binary classification for each of these pairs of coordinates, i.e. to predict whether a given point or pixel belongs to the contour or to the background. In practice, the pairs of coordinates are fed by batches in random order without regard to their respective positions in the 2D space of the original image, i.e. a random flattening procedure. Only after a whole epoch has the FCNN seen all the points (coordinate pairs) that were selected from the image.
We use {\tt fastai} \cite{fastai} {\tt TabularLearner} with a one-cycle policy~\cite{leslie_smith}, maximum learning rate of $5 \times 10^{-3}$, batch size of 8,000 (or less if there are fewer points in the input data) using a training/validation split of $80\%/20\%$ for 300 epochs, at which point the FCNN has typically reached a validation accuracy of $99.9\%$.
COMMENTS FROM REFEREES
*As usual with these kind of studies, it would be good to have some reassurance about the potential systematic errors in the procedure. How strongly do the results depend on tuning some of the hyper-parameters? There are currently a few comments at the end of section II in the manuscript, but it would be useful to expand and provide some more quantitative information. For instance when saying that 'the CNN [...] does not manage to reach the same accuracy...', can this be quantified with a few examples? This point in particular could be important, since the authors suggest that a 'less-perfect' learning would rely on symmetries to encode the image more than a perfect reproduction. This is obviously a very interesting suggestion, which deserves more detailed scrutiny. Surely, if the training is relaxed too much, then informations about the image will be lost. There must be an ideal window where the system works best. Is it possible to explore this feature quantitatively?
TEXT ADDED:
We checked how our results were affected by varying the hyperparameters of the FCNN, in particular as they allow a more precise or looser fit. The most obvious issue is when the FCNNs do not perform so well at their binary classification task: the CNN then has a hard time reaching a good accuracy. For instance, with the same hyperparameters, but only 100 epochs instead of 300, the FCNNs typically reach an accuracy below 99.8\% on the whole training plus validation dataset. CNNs trained on the resulting PCAs barely reach 60\% accuracy on the 5-class task.
Perhaps less obvious is the fact that it is counterproductive to train the FCNNs until they learn their task perfectly: we have tested this by training FCNNs with the same hyperparameters (including in particular 300 epochs), but without defining a validation set. The best model is then selected by minimizing the (training) loss instead of the validation error as was the case before. Such models typically reach an average error rate below 1/10,000 on the whole dataset, i.e. so small that most of them did not make a single mistake in their binary classification task (involving several thousands of points). Here again, the CNNs trained on the resulting PCAs barely reach 60\% accuracy on the 5-class task, possibly because the FCNN has overfit to the location of the individual pixels that have been selected in the random-ssampling process instead of relying on simplifying assumptions such as the symmetry.
COMMENTS FROM REFEREES
*In section II.2, the authors are using ResNet18 for the classification of the symmetries. I wondered if smaller network options have been investigated and if the reason for choosing such an extensive network is purely for increasing accuracy.
TEXT ADDED
Looking for a good accuracy in our image classification task, we perform transfer learning using a Resnet network, but select the smallest one (i.e. ResNet18~\cite{resnet18}) for speed, as implemented in the {\tt fastai} package~\cite{fastai}. From a training sample of 1240 PCAs, we achieve a validation accuracy of $73\%$ on the 5-class problem, and an 80\%-95\% accuracy for each of the four different binary problems of identifying the presence or absence of each symmetry separately.
image.gif
COMMENTS FROM REFEREES
*Can the authors clarify the origin of the error bars of the symmetry bins in figures 4 to 8? Does it appear as a result of running the algorithm many times, as described on page 5? Please correct me if I am missing something, but a neural network will result with the same output every time unless it has a Bayesian layer. Is this because the authors are using multiple distorted template images to classify one symmetry; hence, a class is decided by the statistical significance of this collection?
TEXT ADDED
For each painting, we train the same FCNN architecture several times with the same hyperparameters but using different random initializations of the parameters, yielding different outputs from the hidden layer, and thus different PCAs. We then feed each of these PCAs from the same paintings to the same CNN model we trained on the PCAs of potentials. For each painting, we depict the predicted likelihood for all five symmetries (none $\emptyset$, rotation $O(2)$, continuous translation $T$, discrete translation $T_n$ and reflection $Z_2$) and draw a 68\% confidence interval computed via a non-parametric bootstrap on the predictions obtained from running the same CNN on the PCAs resulting from multiple FCNN runs on the same painting.
COMMENTS FROM REFEREES
*I'm aware that it is hard to release a public code, but if it is possible, can authors also release the analysis code that has been used for this study. I'm sure that the community will appreciate and further develop related studies using their code.
ANSWER
We will be placing our code on Github at the following URL: https://github.com/johanneshirn/symmetry_meets_AI

---

## Round 2 · List of Changes

Some text has been added and modified following the reviewers' suggestions. The details are given above, in the 'Author comments'.

---

## Editorial Decision

published